# Chimeric Antigen Receptor T-Cell Therapy for Colorectal Cancer

**DOI:** 10.3390/jcm9010182

**Published:** 2020-01-09

**Authors:** Daniel Sur, Andrei Havasi, Calin Cainap, Gabriel Samasca, Claudia Burz, Ovidiu Balacescu, Iulia Lupan, Diana Deleanu, Alexandru Irimie

**Affiliations:** 111th Department of Medical Oncology, University of Medicine and Pharmacy “Iuliu Hatieganu”, 400015 Cluj-Napoca, Romania; dr.geni@yahoo.co.uk (D.S.); calincainap2015@gmail.com (C.C.); obalacescu@yahoo.com (O.B.); 2Department of Medical Oncology, The Oncology Institute “Prof. Dr. Ion Chiricuta”, 400015 Cluj-Napoca, Romania; havasi.andrei@gmail.com (A.H.); cburz@yahoo.fr (C.B.); 3Department of Immunology and Allergology, University of Medicine and Pharmacy “Iuliu Hatieganu”, 400162 Cluj-Napoca, Romania; deleanudiana@yahoo.com; 4Department of Functional Genomics, Proteomics and Experimental Pathology, The Oncology Institute “Prof. Dr. Ion Chiricuta”, 400015 Cluj-Napoca, Romania; 5Department of Molecular Biology and Biotehnology, Babeș-Bolyai University, 400084 Cluj-Napoca, Romania; 611th Department of Oncological Surgery and Gynecological Oncology, “IuliuHatieganu” University of Medicine and Pharmacy, 400015 Cluj-Napoca, Romania; airimie@umfcluj.ro; 7Department of Surgery, The Oncology Institute “Prof. Dr. Ion Chiricuta”, 400015 Cluj-Napoca, Romania

**Keywords:** Chimeric antigen receptor (CAR)T-cell, colorectal cancer, immunotherapy, toxicity, trials

## Abstract

Chimeric antigen receptor (CAR) T-cell therapy represents a new genetically engineered method of immunotherapy for cancer. The patient’s T-cells are modified to express a specific receptor that sticks to the tumor antigen. This modified cell is then reintroduced into the patient’s body to fight the resilient cancer cells. After exhibiting positive results in hematological malignancies, this therapy is being proposed for solid tumors like colorectal cancer. The clinical data of CAR T-cell therapy in colorectal cancer is rather scarce. In this review, we summarize the current state of knowledge, challenges, and future perspectives of CAR T-cell therapy in colorectal cancer. A total of 22 articles were included in this review. Eligible studies were selected and reviewed by two researchers from 49 articles found on Pubmed, Web of Science, and clinicaltrials.gov. This therapy, at the moment, provides modest benefits in solid tumors. Not taking into consideration the high manufacturing and retail prices, there are still limitations like increased toxicities, relapses, and unfavorable tumor microenvironment for CAR T-cell therapy in colorectal cancer.

## 1. Introduction

Colorectal cancer (CRC) is one of the most common cancers in 2019 and ranks second for global cancer-related deaths [1]. The prognostic for advanced and metastatic disease is still modest. Approximately one-third of patients are diagnosed with metastatic disease [2]. The median overall survival (OS) with metastasis is about 30 months [3]. Chemotherapy combinations can prevent metastasis and improve OS in first-line treatment of CRC patients [4,5,6]. Despite having multiple lines of treatment for metastatic disease, OS remains low and decreases substantially with time. The addition of targeted therapies achieved a better clinical outcome for these patients. Fluoropyrimidinedoublet (FOLFOX/CAPOX or FOLFIRI/CAPIRI) associated with biologic agents targeting the epidermal growth factor receptor (EGFR) for RAS wild-type tumors or angiogenesis (VEGF) represent the backbone of first and second-line treatment schedule. Targeted therapies such as cetuximab and panitumumab for RAS wild-type patients or antiangiogenic drugs like bevacizumaborziv-afliberceptare the mainstay of metastatic colorectal treatment [7]. The real struggle for clinicians is to find the right balance between standard chemotherapy and new options. Finding the correct management with limited toxicities and increased quality of life and OS is the goal.

A more accurate understanding of the interaction between the immune system and tumor cells has changed therapeutic guidelines by developing new drugs. Immunotherapy with anti-PD-1 mAbs (monoclonal antibodies) pembrolizumab and nivolumab, and anti-CTLA-4 mAbs like ipilimumab have shown promising results in metastatic CRC [8] and are US Food and Drug Administration (FDA) approved for microsatellite instability-high (MSI-H) CRC [9]. The combination of nivolumab and ipilimumab also seems to improve OS and progression-free survival (PFS) in MSI-H metastatic CRC patients and has an acceptable safety profile [10]. Immunotherapy seems to be less effective in CRC compared with other tumor localizations, especially in the mismatch repair (MMR) proficient phenotype and microsatellite stable (MSS) profile [11].

Even after current treatment strategies with chemotherapy, targeted therapies, and immunotherapies, CRC patients develop recurrent disease [12]. Scientists are trying to develop stratification methods and novel treatments for CRC patients. In addition to ongoing clinical trials [9] there are new experimental options. Research in miRNAs [13] and exosomal miRNAs [14] has been promising in the last few years in CRC research. Regarding a CRC vaccination [15], the need for individualization and organized vaccination strategies are still a working process.

Chimeric antigen receptor (CAR) T-cell immunotherapy has become more popular in the last decade in the war against cancer. CARs are laboratory made immune-receptors that modify lymphocytes to target and eliminate cells that express a specific antigen on their surface. T-cells harvested from the patient’s own blood (autologous) or healthy donor’s blood (allogeneic) are genetically engineered to express a specific CAR. For safety reasons, CAR T-cells are conceived to target a specific antigen for the tumor cell and not the normal cell [16]. We investigated the role of CAR T-cells in CRC. We briefly present the main mechanism of action of CAR T-cells, toxicities and administration problems, and implications for other solid tumors. In this review, we focus on literature data to understand if CAR T-cell therapy has a place in the therapeutic sequences of CRC. Data that we present herein confirms that CAR T-cell therapy is a viable method for CRC treatment with the right antigen selection and a combinatorial therapeutic approach.

## 2. Search Criteria

Pubmed, Web of Science, and clinicaltrials.gov were searched with the MeSH terms and keywords chimeric antigen receptor T-cell and colorectal cancer. All the studies that matched were included through August 2019. By reviewing the titles and abstracts, the preliminary screening process identified 49 possible relevant publications. Two separate researchers double-checked the studies included in this review. After eliminating duplicates, other topic articles, non-research work, non-English written papers, and uncompleted reports, 22 articles were found to be relevant to CAR T-cell therapy in CRC.

## 3. Overview and Mechanism of Action of CAR T-Cells

Although CAR T-cell technology was described more than twenty years ago by Gross and colleagues [17], clinical implementation came rather recently. The main interest of CAR T-cell research was to find an active function of T lymphocytes targeting and destroying cancer cells [18]. In recent years, CAR T-cell therapy has come a long way as a personalized immunotherapy option. CAR T-cell therapy trials have achieved long term remissions and complete responses in cancer patients [19]. The process of CAR T-cell therapy is shown in Figure 1.

CAR T-cell therapy manipulates the immune system by collecting and using immune cells to treat cancer. The T-cell is genetically engineered to express special chimeric immunoreceptors (CARs) that give T-cells the ability to target a specific protein. For example, CD19 CAR T-cells are designed to develop CARs against CD19 antigen has a role both in lymphoma and leukemia [20]. Unlike the classical T-cell receptor design that has dual intracellular alpha and beta chains linked with a CD3 complex and associates with the major histocompatibility complex, the CAR design also has an extracellular domain consisting of a single-chain variable fragment (scFv) and does not need the MHC to target specific tumor antigens [21]. 

CAR T-cell therapy is a personalized immunotherapy method that has to follow certain steps to be used in the clinic. After the patient’s immune cells are engineered in vitro, they must be re-infused to complete the process. More descriptively, the whole process consists of the extraction of normal T-cells from the peripheral blood of the patient by leukapheresis, and then CARs are integrated into the T-cells in the laboratory. Afterward follows an in vitro process of cultivation and expansion of the CAR T-cells. The final actions consist of the reinfusion of the final product of CAR T-cells back into the patient and the close monitoring for acute side effects [22]. After reinfusion, the engineered CAR T-cells proliferate and start a killing spree directed towards the tumor cells that bear the specific antigen the CARs are directed against [23]. To optimize efficacy and to minimize toxicities, current protocols suggest conditioning chemotherapy [24].

Considering their structure and production, we are currently in the fourth evolving generation of CAR T-cells. The first generation expressed only the CD3ζ molecule for signaling and was unsuccessful in clinical trials [25]. The second and the third generation had co-stimulatory molecules like CD28, 4-1BB, OX40, or CD27 in addition to CD3ζ to boost cytokines. The fourth generation of CAR T-cells is being designed to overcome the inhibitory effect of the tumoral microenvironment. Having the additional property of secreting cytokines like IL-2 and IL-12, it seems to have a better outcome in solid tumors [26].

There isan increasing number of clinical trials involving CAR T-cell therapy for breast, colorectal, prostate, and renal cancer [27]. Despite the high interest in this treatment, a significant limitation remains accessibility and cost. In 2017, the FDA approved CAR T-cell therapy for several hematological malignancies. Axicabtageneciloleucel (Yescarta) [28] was approved for diffuse large B-cell lymphoma in the context of refractory or relapsed disease. It was also approved in the context of high grade or mediastinal large B-cell lymphoma. Tisagenlecleucel (Kymriah) [29], a CD19 directed 4-1BB/CD3ζ CAR T-cell therapy, was approved rapidly for refractory and relapsed acute lymphoblastic leukemia for children and young adults. Later it was also approved for diffuse large B-cell lymphoma and high-grade B-cell lymphoma. With the hope of finding suitable target antigens for expanding CAR T-cell therapy to other hematological malignancies and solid tumors, many large clinical trials are needed.

## 4. Current Toxicities and Administration Difficulties of CAR T-cells

It is now well established that one of the most significant problems of CAR T-cell therapy is toxicity. There is a clear difference between the toxicity that is encountered in liquid tumors compared to solid tumors. The outcome after treatment with CAR T-cells for solid tumors is still modest. The main side effect of CAR T-cell therapy in hematological tumors is cytokine release syndrome (CRS) [30]. There are also other toxicities like neurotoxicity, anaphylaxis, and on target/off tumor effects [31].

For cytokine release syndrome (CRS), there are a vast number of clinical symptoms. It is not uncommon for the patients to face nausea, malaise, fevers, cardiac dysfunction, hypotension or tachycardia, disseminated intravascular coagulation, or respiratory/renal impairment [32]. This syndrome correlates with CAR T-cell dose, response to therapy, and disease burden [33,34]. CRS can be explained as a response to the torrent of inflammatory cytokines released by the numerous activated CAR T-cells. As cytokines are challenging to evaluate, C-reactive protein (CRP) is used as a marker to assess the severity of CRS [35]. Treatment consists of administration in the first instance with high-dose steroids, vasopressors, respiratory support, and supportive care. Tocilizumab, which is an anti-IL-6R antibody, can be a beneficial treatment for some cases [36].

Neurotoxicity is generally reversible and consists of symptoms of confusion, aphasia, seizures, and delirium. Even though the cause of neurotoxicity is unknown, there are two conjectures about this toxicity in CAR T-cell therapy. The first is related to the high levels of cytokines liberated, and the second to direct injury of the neurological tissue [37,38]. 

On target/off tumor effects that appears more often in the treatment of solid tumors are caused by the presence of the target antigen for CAR T-cells on cancer cells and also in low levels in normal tissue [39]. CAR T-cells acting on normal cells cause this sort of toxicity, underscoring the importance of choosing the right target antigen for therapy.

A recent paper by Shah and Fry [40] described the barriers that CAR T-cell therapy faces to induce long-term remission. The first of the four barriers is the failure to achieve remission either by problems in manufacturing, infusion, or CAR T-cell activation and expansion. Lack of access is an issue, as is the cost of treatment. The second barrier leading to disease relapse is constituted by antigen modulation, which enables antigen escape responsible for resistance. This phenomenon is also found in solid tumors. The third barrier is the toxicity of CAR T-cell therapy including cytokine release syndrome or neurotoxicity. Finally, there are issues of adapting CAR T-cell therapy for other types of hematological malignancies or solid tumors. For instance, anti-CD19 CAR T-cell therapy for adults with lymphoma is correlated with a good response compared with the clinical response of CAR T-cell therapy in leukemia and pediatric malignancies. 

Of course, there are a series of limitations to CAR T-cell therapy like tumor type, patient population, lymphodepletion regimen, and CAR T-cell design. Developing guidelines [41] and having an early intervention concerning toxicity using adequate grading scales [42] can improve the results of this promising method. However, data from the literature reports one severe event of a patient with metastatic CRC that died after receiving Trastuzumab CAR T-cell therapy associated with IL-2 after lymphodepletion chemotherapy. The cause of death was rapid respiratory failure onset due to reactivity against pulmonary tissue [43]. 

Suicide genes or elimination genes come as a solution for these sorts of toxicities. Incorporating genes like Herpes simplex virus thymidine kinase or Escherichia coli nitroreductase have some downfalls. Introducing these systems into limited genetic space can cause immunogenic effects resulting in premature elimination of CAR T-cells and subsequent toxicity [44]. Although suicide gene engineering is part of the current clinical trials, proper procedures have not been established in clinical practice.

## 5. CAR T-Cell Therapy in Solid Tumors

CAR T-cell therapy is increasingly becoming a key player in the treatment of malignant diseases. While clinical trials are still in high demand, the use of CAR T-cells in the therapy of hematologic oncology such as acute lymphoblastic leukemia or lymphoma is generally accepted worldwide [45]. By targeting CD19 antigen on B-cells using CAR T-cell therapy, clinicians attained durable remissions and clinical response in patients with B-cell non-Hodgkin lymphoma and acute lymphoblastic lymphoma [46]. Efforts are being made to provide CAR T-cell treatment for other types of lymphoma and myeloma. 

Most published work has mainly focused on hematologic disease, and results concerning solid tumors have insufficient evidence for clinical use. There are various challenges for CAR T-cell therapy in solid tumors. 

Recent trials moved their attention to solid tumors by targeting different surface proteins. From the multitude of targeted surface proteins like human epidermal growth factor receptor 2 (HER2), diganglioside (GD2), carcinoembryonic antigen (CEA), mesothelin, and fibroblast activation protein (FAP), only the first two (GD2 and HER2) were part of positive trials [47]. Only one trial using GD2 CARs for neuroblastoma resulted in complete remission [48]. Using HER 2 CARs for sarcoma or HER1 for advanced relapsed or refractory non-small cell lung cancer showed stable disease or partial response in the patients enrolled [49,50]. 

Several biological barriers exist in the process of successful trafficking of T-cells from blood to the stromal elements of solid tumors. These barriers could explain the different results of CAR T-cell therapy in hematological malignancies compared to solid tumors. Even if the prerequisite of trafficking and infiltration is done, T-cells face some natural obstacles and become defective. One of the most critical features is that CAR T-cells first have to identify the proper tumor-associated antigen (TAA). Then they have to face the unwelcoming tumor microenvironment. The tumor microenvironment interferes with the penetration of CAR T-cells into stromal tumor mass, promoting an immunosuppressive milieu [51]. Afterward, CAR T-cells encounter cytokines and soluble inhibitory factors. Suppressive immune cells like regulatory T-cells (Treg) or tumor-associated macrophages (TAM) and other negative regulatory systems also step in the way of the activated T-cells [52,53]. 

Researchers try to understand the interactions between the hostile microenvironment and the host’s immune system with all its immunomodulatory effects. These two seem to be the culprits for the resistance mechanisms of CAR T-cell therapy in solid tumors. By changing the design of CAR T-cells incorporating co-stimulatory molecules, ligands, targeted therapies, or immunomodulatory agents can improve the clinical outcome [54]. Wang et al. [55] suggest that targeting more than one antigen and building a dual-targeted CAR can overcome the risk of antigen escape relapse and diminish the effect of antigen heterogeneity. Off-target toxicity is still a big issue to consider in classically designed CAR T-cells and in novel dual-targeted cells [56]. PD-1/PD-L1 blockade may be a successful plan for boosting the potency of CAR T-cell therapies in solid tumors [57]. 

Recent data assume that new mechanisms of antigen recognition through CARs can make a difference in solid tumors. Trying to modulate the immunological synapse formation can improve efficacy. The density of the target antigen can also control the efficacy of treatment in solid tumors [58].

Personalized modification of CAR T-cell construction, finding a more suitable T-cell subset for engineering, incorporating anti-cancer cytokines, manipulating PD1/CTLA4 checkpoints, infiltrating soluble immunosuppressive factors like TGF-β and IL10, are some proposed changes that can make a difference [59]. An opportunity for modifying CAR design and construction for use on other cells than immune cells like the design of a tumor cell vaccine exists as a promising research direction [60]. 

Many experts now contend that we should change the general approach of CAR-T cell therapy in solid tumors. We should focus more on CAR-T local delivery and on associating chemotherapy or immune checkpoint inhibitors with the regimen for metastatic disease. Targeting two different antigens seems a reliable solution to broadening the spectrum of therapeutics. Scientists should consider developing CAR T-cell therapy for adjuvant treatment after resection to prevent recurrence or metastasis [61].

## 6. CAR T-Cell Therapy in Gastrointestinal Malignancies

Gastrointestinal tumors are the most common types of human cancers encountered worldwide. There is a modest improvement in survival by integrating novel chemotherapeutic and radiotherapeutic schedules. Due to the increasing burden of cancer, it is necessary to incorporate alternative strategies like CAR T-cell therapy to improve outcomes. CAR T-cell therapy has shown modest benefits in solid tumors. There are several attempts to find the right formula for CAR T-cells in gastrointestinal tumors. 

### 6.1. Pancreatic Cancer

CAR T-cell therapy is studied intensively in pancreatic cancer. Like most solid tumors, pancreatic cancer presents a series of tumor-specific antigens that can serve as promising targets for CAR T-cell therapy. Overexpressed antigens like carcinoembryonic antigen, mesothelin, human epidermal growth factor receptor 2 (HER-2), and mucin 1 (MUC1) show great potential [62,63,64]. With this in mind, an article from 2017 [65] summarized the possible targets of CAR T-cells in pancreatic cancer and added CD24, prostate stem cell antigen, and natural killer receptors for the construction of CARs. By showing high expression, mesothelin appears to be the most promising target for CARs [66,67,68,69]. Other research [70] focused on blocking IL-10 with the purpose of tackling the immunosuppressive effect of the tumor microenvironment and enhancing mesothelin CAR T-cell activity. A phase I trial showed good results using mesothelin mRNA CAR T-cell therapy against pancreatic cancer metastases. The trial also revealed a satisfactory safety profile [71]. In lung metastases from pancreatic cancer, mesothelin CAR T-cell design showed impressive results [72]. Combining mesothelial CAR T-cells with cytokine-armed oncolytic adenovirus expressing TNF-α or IL-2 enhanced the efficacy of the chimeric antigen receptor therapy [73].

Another approach was targeting prostate stem cell antigen and protecting cells from immunosuppressive cytokines by manipulating the tumor microenvironment [74]. Abate-Daga and colleagues [75] proposed targeting prostate stem cell antigen in a humanized mouse model of pancreatic cancer. Modulating the dense stromal surroundings of pancreatic cancer and adapting the cells to the tumor microenvironment seems like proper options to improve clinical benefit [76]. With a different CAR construct, humanized CD47-CAR T-cells eliminated with excellent specificity ovarian, cervical, and pancreatic cancer cells [77]. Zhang E et al. [78] constructed a dual CAR-modified T-cell to eradicate AsPC-1 pancreatic cells that have high expression of carcinoembryonicantigen (CEA) and mesothelin (MSLN). A study using switchable CAR T-cells to target HER-2 on patient-derived xenograft models from patients with aggressive metastatic pancreatic cancer showed considerable potency [79]. Whilding et al. hypothesized that controlling IL-8 might control tumor burden in solid tumors by accomplishing a higher therapeutic activity in pancreatic and ovarian tumor xenografts. This result was possible by expressing IL-8 receptors CXCR1 and CXCR2 in a CAR T-cell construct to target the tumor-associated αvβ6 integrin [80]. 

These findings mentioned above confirm that CAR T-cell therapy for pancreatic cancer is developing fast, and its potential needs to be confirmed in future trials and studies [81,82].

### 6.2. Hepatocellular Carcinoma

Accumulating data show that hepatocellular carcinoma has limited therapeutic options. Hoseini et al. outline the possibility of using CAR T-cell therapy and bispecific antibodies for hepatocellular carcinoma treatment. Bispecific antibodies can redirect natural killer cells toward cancer cells paving the way for CAR T-cells to infiltrate the tumor site [83]. Although no clinical CAR T-cell trials for hepatocellular cancer have been completed to date, some preclinical and clinical evidence suggests a potent antitumor activity for therapies targeting CEA, MUC-1, and GPC-3 antigens [84,85]. Most results in experimental studies have come from targeting GPC-3, either using an inducible armored IL-12 construct [86] or by directly eliminating GPC-3 positive HCC cells [87,88]. Another approach was by a dual-CAR construct expressing GPC-3 and asialoglycoprotein receptor 1 (ASGR1) with less toxicity [89]. One study [90] proved in vitro and in vivo that by disrupting programmed death 1 receptor (PD-1) on the Glypican-3 (GPC3)-targeted second-generation CAR T-cells showed a much stronger activity against hepatocellular carcinoma. 

For immunotherapy to be effective in hepatocellular carcinoma, it needs a more combinatorial perspective by associating checkpoint inhibitors with immune cell therapy [91,92]. There is still a long way for CAR T-cell therapy in HCC from bench to bedside. 

### 6.3. Gastric Cancer

There is little evidence about CAR T-cell therapy in gastric cancer. The data available is rather recent proposes several targets for CARs. The first study showed that the construction of a CAR T-cell that targets monoclonal antibody 3H11 had a promising response in gastric cancer, although it does not overcome the current biological barriers of solid tumors [93]. The second [94] study used the fact that folate 1 receptor (FOLR1) expression is overexpressed in gastric cancer in comparison to normal tissue. The results confirmed that FOLR1 CAR T-cells could recognize and exhibit anti-tumor activity in FOLR1-positive gastric cancer cells. The third study chose natural killer group 2D(NKG2D) ligand as a target for a second-generation CAR design. The final NKG2D-CAR T-cells compound showed cytolytic activity against gastric cancer cells. Adding cisplatin increased the responsiveness to the CAR T-cell construct [95]. The latest research concerning gastric cancer involved targeting claudin18.2 for their CAR T-cell construction [96]. 

### 6.4. Esophageal Cancer

Clinical trials of CAR T-cell therapy in thoracic malignancies are scarce. The main targets for CARs of esophageal cancer were epithelial cell adhesion molecule (EpCAM) and HER2 [97]. We found one experimental study that targeted EphA2 for esophageal squamous cell carcinoma (ESCC) CAR T-cell construction. The EphA2 CAR T-cells showed a better esophageal cell kill ratio than T-cells and were also dose-dependent [98]. 

### 6.5. Biliary Tract Cancer

Immunotherapy has shown clinical benefit in hepatobiliary cancer but only in a small subset of patients. Recently investigators have examined the possible implication of CAR T-cells in biliary tract cancer [99]. CAR T-cell therapies studies and trials in biliary tract cancers propose mesothelin [100], EGFR [101,102], or HER2 [103] as targets because of their tendency to be overexpressed in these malignancies.

## 7. CAR T-Cell Therapy Studies for CRC

Potential targets for CAR T-cell therapy in CRC are shown in Figure 2. Hege et al. [104] reported one of the first human trials concerning CAR T-cells for metastatic CRC. It consisted of two phase 1 trials (C9701 and C9702), which used the same CART72 cells. CART72 cells were designed as first generation CAR T-cells that targeted tumor-associated glycoprotein (TAG)-72 and included a CD3-zeta intracellular signaling domain. The difference between the two trials was the administration of CART72. Trial C9701 had the CART72 administrated intravenously in an escalating dose, and trial C9702 had direct hepatic artery infusion in patients with colorectal liver metastases. The results showed a good safety profile despite a short-term persistence in blood, and the fact that trafficking to tumor tissue was limited. In addition, CART72 immunogenicity was associated with fast clearance after infusion of CAR-T cells. 

Two articles described guanylylcyclase2C (GUCY2C) as a possible target for CARs. The first article [105] provides evidence that GUCY2C CAR T-cells can treat parenchymal CRC metastases in a mouse model without autoimmunity. The second article [106] demonstrated the effectiveness of GUCY2C targeted CAR T-cell therapy against metastatic tumors in mouse models and in xenograft models of human CRC.

Initial studies with chimeric antigen receptor targeting CEA antigen-expressing CRC and its liver metastases indicate specific anti-tumor activity and the probability of avoiding immunosuppression [107]. Using CEA-positive CRC patients, the NCT02349724 Phase I trial developed CEA CAR T-cell therapy. Ten refractory and relapsed patients with metastases were enrolled in this trial. The endpoint showed good tolerability of CEA CAR T-cells even with high doses, and some efficacy in the patients treated [108]. Another approach was using CAR T-cells in a CEA-positive mouse model induced to develop colitis. The results of the study demonstrated that CEA CAR T-cells can ameliorate ulcerative colitis and can delay the transformation to CRC [109]. The combination of CEA CAR T-cells with recombinant human IL-12 has improved anti-tumor activity in colorectal, pancreatic, and gastric cell lines [110]. 

Retroviral genetic transduction of the single-chain variable domain anti-carcinoembryonic Ag (CEA) Fcε receptor I γ-chain fusion (scFv anti-CEA) receptor amplified expression of this receptor in naïve mouse T lymphocytes as shown by Darcy et al. Additionally, they established that the altered T-cells were capable of inducing CEA-positive cell lysis via perforin-mediated pathways and emphasized the importance of interferon γ in colorectal tumor control. Complete tumoral eradication could not be obtained in interferon γ deficient T-cells [111].

Oncoretroviral transfection is exceedingly useful in transducing chimeric T-cell receptor in fast-dividing T-cells, but involves quickly multiplying cells and production of high virus titers. The transfection of peripheral blood lymphocytes (PBL) with retroviruses proved low efficiency. Lentiviruses can transduce non-dividing cells. The association of a lentivirus with vesicular stomatitis virus G protein gave highly efficient PBL transduction of chimeric T-cell receptor. PBL-modified cells proliferated in vitro on exposure to CEA-positive cells and showed an antitumoral effect on CEA-positive CRC mouse models [112]. 

CAR T-cell therapy was associated with anti-4-1BB on human-Her2 mouse models to enhance the anti-tumor activity. The study revealed that anti-4-1BB monoclonal antibody could augment the effect of CAR T-cell therapy by suppressing the host’s tumor microenvironment. This protocol could be of use for solid tumors like CRC after first line adoptive therapy failure [113].

Daly et al. conducted one of the first studies using CAR T-cell therapy for CRC [114]. The study redirected T-cells using a chimeric receptor which recognized epithelial glycoprotein 40(EGP40) as the selected antigen. Resultsof another study showed that MiR-153 has the property to enhance CAR T-cell therapy in CRC. MiR-153 inhibits indoleamine 2,3-dyoxygenase 1(IDO1) in CRC cells and acts as a tumor suppressor [115].

Intraperitoneal infusion of CAR T-cells combined with depleting antibodies against myeloid-derived suppressor cells (MDSC) and regulatory T-cells (Treg) showed considerable results in CRC mouse models. Regional delivery of CAR T-cell compared to systemic infusions resulted in enhanced anti-tumor effects for peritoneal carcinomatosis [116]. For peritoneal carcinomatosis, an EpCAMtargeting CAR T-cell construct exhibited anti-tumor efficacy for mouse models of human ovarian cancer and CRC [117]. EpCAM proved to be a suitable target for CARs in a xenograft mouse model. The EpCAM CAR T-cell delayed tumor growth and displayed an excellent safety profile [118]. Deng et al. [119] showed that NKG2D CAR T-cells exhibit specific cytotoxicity in human CRC cell lines. The study was conducted in vivo and in vitro showing promising immunotherapeutic activity. In a patient-derived xenograft mouse model with CRC, HER-2 CAR T-cell therapy selectively killed HER-2 positive tumor cells. Besides the establishment of a new study model for immunotherapy, the study revealed that the cells were protected from tumor re-challenge after infusion [120]. 

Knowing that the hostile microenvironment is part of the resistance system of CAR T-cell therapy in CRC, a study evaluated the implication of Interferon-alpha/beta receptor alpha chain (IFNAR1). It showed that genetic or pharmacologic stabilization of IFNAR1 can influence tumor growth and can enhance chimeric antigen receptor T-cell therapy in solid tumors [121]. 

One preliminary study that is part of an ongoing clinical trial showed that CD133 could be an interesting target for CAR T-cell therapy. Infusion with CAR T-133 cells for positive CD133 metastatic pancreatic cancer, hepatocellular cancer, and CRC showed effective response with tolerable toxicity. Another practical implication was that by repeating cell infusions, a more extended period of stable disease resulted [122].

From another perspective, the use of antibiotics proved to diminish the effect of CD4+ CAR T-cells in a CRC mouse model. Chemotherapy with cyclophosphamide was used for the conditioning regimen in mice [123]. Using colorectal carcinoma cells positive for EGP-2 researchers proved that there was no significant difference in T-bodies activity with either ζ-chain or the ϒ-chain in chimeric T-cell receptor construction. This confirmed the option of using both ζ-chain orϒ-chain in clinical trials [124]. Sasaki et al. generated gene-modified Tc1 and Th1 cells from one CRC patient. These gene-modified T-cells showed high cytotoxicity and produced INF-γ in the tumors expressing CEA [125].

### CAR T-Cells Ongoing Trials in CRC

Several ongoing trials (Table 1) are investigating the use of CAR T-cell in the treatment of CRC. A phase I, open-label clinical trial is assessing the safety, cellular kinetics, and efficiency of CYAD-101, an allogenic CAR T-cell therapy targeting ligands of NKG2D, administered concurrently with FOLFOX in patients with unresectable metastatic CRC [126]. 

Two clinical trials are researching the efficiency and safety of NKR-2 CAR T-cells in metastatic CRC patients. The first trial with NCT03370198 ID [130] investigates the biweekly hepatic transarterial administration of NKR-2 CAR T-cells in patients with unresectable liver metastases. On the other hand, the second trial [131] studies the safety and efficiency of NKR-2 cells administered alongside FOLFOX in potentially resectable liver metastases from CRC.

EGFR and EGFR IL 12 CAR T-cell safety and feasibility in the treatment of metastatic CRC are also being currently evaluated in phase I and II studies [127,128]. Various trials are investigating the use of anti-carcinoembryonic antigen targeted CAR T-cells in several CEA-positive malignancies, including CRC. Concerning the administration protocol, trials with CEA as a target for CARs prefer systemic, hepatic transarterial administration, vascular interventional, or intraperitoneal infusion. In addition tometastatic colorectal cancer, CEA is used as a target for other solid tumors like lung, gastric, pancreatic, hepatocellular, or breast cancer [129,134,135,139,141]. 

Anti HER2 CAR T-cells are studied in HER2 positive cancers in preclinical studies [133]. NCT03740256 [140] phase 1 trial is investigating the efficiency and safety of HER 2 chimeric antigen receptor-modified adenovirus-specific cytotoxic T lymphocytes administered in association with intratumoral injection of CAdVEC, an oncolytic adenovirus. The administration of CAdVEC is proposed to create a pro-inflammatory tumor microenvironment that will promote the recruitment and expansion of the transferred CAR T-cells. 

The use of CAR T-cells targeted against MUC1 is proposed for relapsed or refractory solid tumors like glioma, metastatic colonic adenocarcinoma, and gastric cancer, which exhibit confirmed MUC1 positive status [132]. C-MET appears as a target for colorectal cancer, hepatoma, ovarian, and renal cancer in phase I/II CAR T-cell trial with over 73 participants. The trial consists of a multi-target gene-modified CAR T/TCR T-cell that includes ten different tumor-specific antibodies [136]. Other ongoing trials use CAR T-cells targeting EpCAM [137], or CD 133 [138] -positive CRC to assess safety and feasibility for both solid tumors like colorectal cancer as well as hematological malignancies. The completion date for the trial concerning CD133 targeted CAR T-cells is December 2019. Preliminary data suggest that CD133 is an attractive therapeutic target for relapsed or refractory malignancies, including CRC. The trial underlines the safety profile and efficient activity of CART-133 therapy for CD133 positive metastatic cancers.

## 8. Conclusions and Further Perspectives

Despite the recent developments in CAR T-cells therapy for hematological malignancies, the use of these therapies in solid tumors is still debatable. CAR T-cell therapy has become mainstream for hematologists even though there are a series of limitations to the treatment. Adoptive cell therapy should be conducted in highly specialized clinical centers with staff that is trained in dealing with common side effects. Product manufacturing and infrastructure for clinical implementation is well accepted in many centers worldwide, but a more specific focus on solid tumors is in need. 

As summarized in this review, there are many promising CAR T-cell therapeutic strategies for colorectal cancer that have shown success in preclinical models or early phase clinical trials. The primary role of immunotherapy as CAR T-cell is confirmed in the durable response of the treatment. One of the major issues is that durable response is available only for a small subset of patients in colorectal cancer. The objective of CAR T-cells therapy is to find the right target for CARs or the right combination with novel checkpoint inhibitors or monoclonal antibodies. This approach can help to broaden the spectrum of patients that can obtain a sustainable clinical benefit. 

Chimeric antigen receptor T-cell therapy is an immunological concept that has been in development for years. Treatment of hematological malignancies with this method has changed the way we treat cancer and makes us see the possibilities of treating solid tumors. Recent data show that CAR T-cell therapy could be useful in gastrointestinal tumors in colorectal, pancreatic, gastric, or hepatobiliary cancer. This has opened the door for further studies and trials, although there is extensive work needed to implement this therapeutic method in colorectal cancer. Colorectal cancer is one of the most studied cancers and one of the few with promising immunotherapy results. CAR T-cell therapy will hopefully deliver substantial clinical benefit for colorectal cancer.

## Figures and Tables

**Figure 1 jcm-09-00182-f001:**
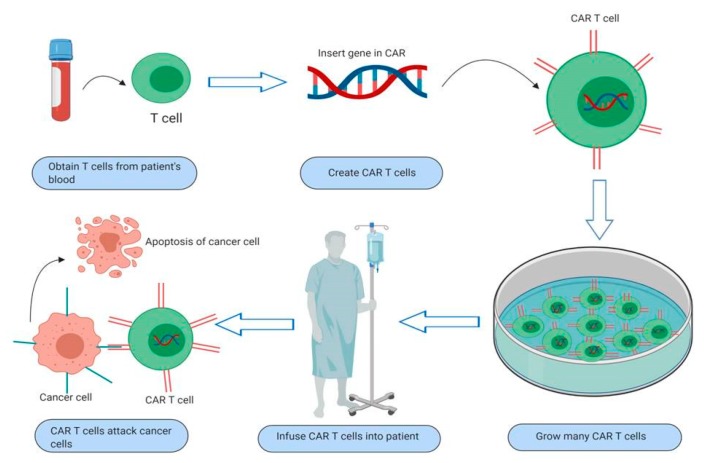
Overview of chimeric antigen receptor (CAR) T-cell therapy. Process of extracting normal T-cells from the patient’s peripheral blood; integration of CARs in T-cells in the laboratory; in vitro cultivation and expansion of CAR T-cells that are re-infused into the patient’s bloodstream; CAR T-cells proliferate and kill the tumor cells that bear the specific antigen the CARs are directed against.

**Figure 2 jcm-09-00182-f002:**
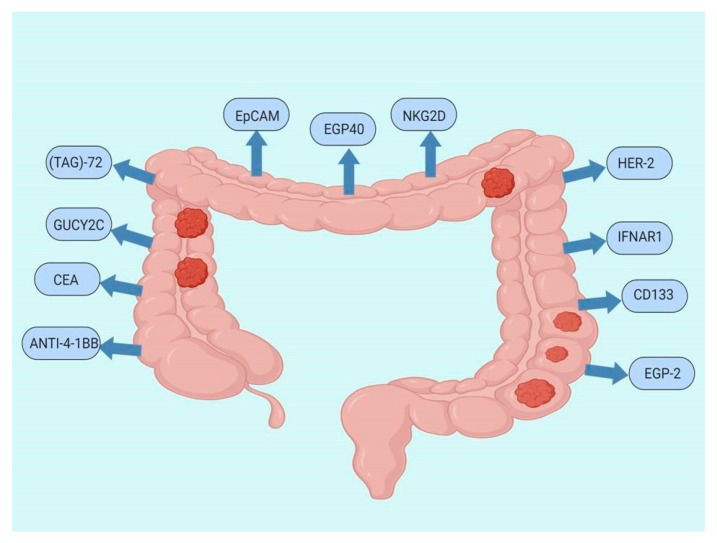
Targets for CAR T-cell therapy in CRC: Anti-4-1BB; CEA; guanylylcyclase2C (GUCY2C); TAG-72; EpCAM; epithelial glycoprotein 40(EGP40); NKG2D; HER-2; interferon alpha and beta receptor subunit 1(IFNAR1); prominin-1 (CD133); epithelial glycoprotein-2 (EGP-2).

**Table 1 jcm-09-00182-t001:** Ongoing clinical trials of CAR T-cell therapy conducted in CRC.

Target	Pathology	Trial ID	Study Phase	Administration	Patient Number	Year	Reference
EGFR IL-12	Metastatic colorectal cancer	NCT03542799	I/II	Systemic	20	2018	[127]
EGFR	EGFR-positive colorectal Cancer	NCT03152435	I/II	Systemic	20	2017	[128]
NKG2D	Metastatic colorectal cancer	NCT03692429	I	Systemic	36	2018	[126]
CEA	Metastatic colorectal cancer	NCT02959151	I/II	Vascular interventional therapy or intratumoral injection	20	2016	[129]
NKR-2	Unresectable liver metastasis of colorectal cancer	NCT03370198	I	Hepatic transarterial	18	2017	[130]
NKR-2	Potentially resectable liver metastasis of colorectal cancer	NCT03310008	I	Systemic	36	2017	[131]
MUC 1	Colorectal cancer	NCT02617134	I/II	Systemic	20	2015	[132]
HER2	Colorectal cancer	NCT02713984	I/II	Systemic	60	2016	[133]
CEA	Colorectal cancer	NCT02349724	I	Systemic	75	2015	[134]
CEA	Peritoneal metastases or malignant ascites of Colorectal cancer	NCT03682744	I	Intraperitoneal infusion	18	2018	[135]
C-MET	Colorectal cancer	NCT03638206	I/II	Systemic	73	2018	[136]
EpCAM	Colorectal cancer	NCT03013712	I/II	Vascular interventional therapy	60	2017	[137]
				Endoscopy mediated infusion			
CD133	Colorectal cancer	NCT02541370	I/II	Systemic	20	2015	[138]
CEA	CEA + liver metastases from gastrointestinal tumors including colorectal cancer	NCT02850536	I	Hepatic transarterial	5	2015	[139]
				Intrapancreatic retrograde venous infusion			
HER2	Colorectal cancer	NCT03740256	I	Systemic &Intratumoral	39	2018	[140]
CEA	CEA + adenocarcinoma with liver metastases from gastrointestinal tumors including colorectal cancer	NCT02416466	I	Hepatic transarterial administration	8	2015	[141]

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
