# Peer review of "Chimeric Antigen Receptor T-Cell Therapy for Colorectal Cancer"

_jcm, 2020, doi:10.3390/jcm9010182_

Round 1
Reviewer 1 Report
The authors provide a comprehensive compilation of CAR T cell therapy, their mechanism of action, associated toxicities and ongoing clinical trials, with specific focus on CRC. They provide literature from 22 relevant publications on CAR T cell therapy in CRC. The review has a logical flow with adequate references. The authors highlight that despite the success of CAR T cell therapy in hematological malignancies, the efficacy in solid tumors is limited, owing to restricted access by stroma, survival in the hostile tumor microenvironment (TME) and suppression by Tregs and MDSC in the TME. Thus, combination strategies to promote T cell infiltration, survival and/or activation in the TME are beneficial. The authors mention few examples of combination therapy such as IL-12, IL-2 as well as depleting antibodies of MDSC and Treg. In addition, the route of delivery also determines the efficacy of CAR T cell therapy. Local intratumoral delivery can overcome tumor-stroma-mediated barrier while intraperitoneal delivery and intravenous delivery have shown promise in mouse models and human clinical trials respectively. In addition, the authors underscore the importance of choosing the right antigen(s) to overcome therapeutic resistance and on-target/off tumor effects and also provide ample examples of antigens that are being used in CAR T cell generation currently. The table summarizing ongoing clinical trials in CRC is well made and easy to follow.
There are minor grammatical errors throughout the text that need review. Also, there is an error in line 200 (page 5) where the authors mention that the TME acts as a physical barrier and prevents an immunosuppressive effect whereas in reality, the TME promotes an immunosuppressive milieu.
With these minor changes incorporated, the review is otherwise well compiled and makes a good contribution to the field.
Author Response
Dear Reviewer 1,
We thank you for your pertinent observations regarding our study and we consider that by addressing your request we improved the quality of our manuscript entitled “ Chimeric Antigen Receptor T-cell Therapy for Colorectal Cancer”.
Request:
There are minor grammatical errors throughout the text that need review. Also, there is an error in line 200 (page 5) where the authors mention that the TME acts as a physical barrier and prevents an immunosuppressive effect whereas in reality, the TME promotes an immunosuppressive milieu.
Answer:
We have accommodated our manuscript correcting it as you mentioned. We double-checked and corrected the text for grammatical errors. We also corrected the error from page 5 after your very well pointed suggestion.
The tumor microenvironment interferes with the penetration of CAR T-cells into stromal tumor mass, promoting an immunosuppressive milieu.

Reviewer 2 Report
The manuscript entitled "Chimeric Antigen Receptor T-cell Therapy for
3 Colorectal Cancer" by Sur et al provides a clinical overview about an innovative immunotherapuetic approach CAR-T based in the CRC patients. The manuscript is well written, but need some minor revisions:
In the introduction section, please, could the authors report the clinical practice adopted to treat CRC patients? In particular, my suggestion is to explain the clinical role of target therapy TKI based that represents the first line approach for these patients. In the introduction section, please, could the authors better clarufy technical process that lead to CAR-T cell preparation? In particular, could the authors define if quality check in CAR T preparations are required ? In each sub section, please could the authors report the preliminary clinical data when they are available?
Author Response
Dear Reviewer 2,
We thank you for your pertinent observations regarding our study and we consider that by addressing your request we improved the quality of our manuscript entitled “Chimeric Antigen Receptor T-cell Therapy for Colorectal Cancer”.
Request 1:
In the introduction section, please, could the authors report the clinical practice adopted to treat CRC patients? In particular, my suggestion is to explain the clinical role of target therapy TKI based that represents the first line approach for these patients.
Answer:
We have accommodated our manuscript correcting it as you mentioned.
Fluoropyrimidine doublet (FOLFOX/CAPOX or FOLFIRI/CAPIRI) associated with biologic agents targeting the epidermal growth factor receptor (EGFR) for RAS wild-type tumors or angiogenesis (VEGF) represent the backbone of first and second-line treatment schedule. Targeted therapies such as cetuximab and panitumumab for RAS wild-type patients or antiangiogenic drugs like bevacizumab or ziv-aflibercept are the mainstay of metastatic colorectal treatment.
Request 2:
In the introduction section, please, could the authors better clarify technical process that lead to CAR T-cell preparation? In particular, could the authors define if quality check in CAR T preparation are required?
Answer:
CARs are laboratory made immune-receptors that modify lymphocytes to target and eliminate cells that express a specific antigen on their surface. T cells harvested from the patient’s own blood (autologous) or healthy donor’s blood (allogeneic) are genetically engineered to express a specific CAR. For safety reasons, CAR T-cells are conceived to target a specific antigen for the tumor cell and not the normal cell.
*As far as we know, there is no advised quality check for CAR T-cell preparation. The only method that can be done for the moment to bring a cellular therapy like this with a complex manufacturing process to a larger, international patient population is to generate high-quality vectors and to understand the long-term safety of gene therapy and anticipate global regulatory concerns.
Request 3:
In each sub section, please could the authors report the preliminary clinical data when they are available?
Answer:
We revised the status of all the trials and studies that promoted the use of specific targets for CAR T-cell therapy and we couldn’t find significant updates at the moment. For the ongoing clinical trials, the patients recruitment status wasn’t reached considering most of the trials are relatively recently developed. Some of the studies have completion deadline for 2021-2023.
*The clinical trial relevant for ref. 141 finished on 17 Dec 2019 and has determined the safety and feasibility profile of the chimeric antigen receptor T cells transduced with the anti-CD133 for relapsed and refractory malignancies including CRC. CD133 acts as an attractive target for CAR T-cell therapy.
